# Environmental Accounting Information Disclosure Driving Factors: The Case of Listed Firms in China

**Maoli Ji [1], Yuguang Ji [2],\* and Shulan Dong [1]**

1   School of Business, Jiaxing University, Jiaxing 314001, China
2   School of Business and Management, Jilin University, Changchun 130015, China
\*   Correspondence: jiyg21@mails.jlu.edu.cn; Tel.: +86-155-2683-2070

**Abstract:** This study explores factors that drive environmental accounting information disclosure (EAID) among corporations in China. Using a sample of 200 A-shared listed firms, we apply a structural equation model (SEM) and multiple linear regressions to examine how, and to what extent, external pressure, corporate performance and corporate governance affects the EAID of corporations. The results show that external pressure and corporate performance can significantly and positively affect corporate EAID. Regarding external pressure, government regulations, media pressure and loans are the most important driving factors, whereas profitability and sales ability are the most important ones among corporate performance factors. However, we found that governance factors have no significant impact on EAID. This paper enriches research on environmental accounting information disclosure and provides important insights for Chinese regulators into effective ways of fostering disclosures of environmental accounting information and raising corporate awareness of CSR fulfillment to ensure sustainable development.

**Keywords:** external pressure; corporate performance; corporate governance; environmental accounting information disclosure; CSR; structural equation model

## 1. Introduction

Since the Chinese government proposed that "China strives to peak carbon dioxide emissions by 2030 and achieve carbon neutrality by 2060" at the 75th Session of the United Nations General Assembly in 2020, "Peak carbon dioxide emissions" and "carbon neutrality", abbreviated to "Double-Carbon" strategy, has become "Word of the Year" in many fields, such as energy and technology, in China. To realize this strategy, attention should be paid to the main bodies responsible for carbon emissions, the corporations. A corporation reflects its situation on all aspects to the outside world through various kinds of information and, by doing this, is then subject to the supervision and evaluation of all parties. Amongst the information disclosed by corporations, environmental accounting information is the main means to convey social responsibility performance to the government and public.

However, currently, there are no mandatory laws forcing China's corporations to disclose environmental accounting information, which means that environmental accounting information is "voluntary information". Under such circumstances, corporations may view environmental accounting information as "additional information" and lack the "impetus" to disclose the pertinent information. Although the Listing Rules of Shanghai Stock Exchange (Revised in January 2022) require that "Listed firms should compile and disclose their CSR information", they do not stipulate the detailed requirements and corresponding punishments for those who do not obey. Most corporations still consider it a "twice-told" "ornamental" rule and do not attach enough importance to it. In recent years in China, existing evidence indicates that many problems have already been exposed in the field of corporations' environmental accounting information disclosure (hereinafter abbreviated as "EAID"), such as disclosure information containing too much qualitative information with

strong subjectivity and inferior quality. As adequate disclosure by corporations of their own environmental accounting information is a prerequisite for environmental governance [1], only by improving the quality of corporate environmental accounting information disclosure can China's government build a solid foundation for the realization of the "Double Carbon" national strategy. Therefore, it is of great significance to explore factors influencing and driving China's corporations to disclose their environmental accounting information.

Previous studies have explored the relationship between one or two external pressures [2–7], corporate performance [8–12], corporate governance [13–17] and corporate environmental accounting information disclosure through empirical analysis or other methods, but no literature has aggregated these factors together and considered the configuration relationships to explore the relationship between them and EAID. In addition, from the perspective of methods, most previous studies used regression analysis. The paper here presented represents the first time SEM (structural equation model) is utilized to explore this field. This paper opted to use SEM for the following reasons.

Firstly, potential variables can be better measured using SEM. For each potential variable, there are four observed variables to measure it. Take "external pressure" as one example. In this paper, the following four observed variables were allocated to the latent variable "external pressure", in regard to creditors, government and the public: "bank loans", "state-owned equity proportion", "media pressure of public opinion", "whether audited by the Big Four", respectively. By doing this, contingency is reduced, and, to some extent, credibility of the conclusion assured.

Secondly, configuration interaction between explanatory variables can be considered while exploring the relationship between explanatory variables and explained variables. Take corporate performance and corporate governance as an example. Much previous literature showed that the ownership structure of a company significantly affects corporate performance. For example, Hu (2020) [18] believed that there was a "U-shaped" relationship between corporate performance of listed companies and the proportion of state-owned shares. Coincidentally, Lang L.H.P. (1994) [19] found a significant positive correlation between equity balance and return on equity (ROE) through empirical research. However, traditional quantitative regression analysis takes an isolated analysis perspective between independent variables and dependent variables, and it is difficult to address the complex inner relationship of how interdependence of variables and their "configuration effect" influence the results with such an approach [20]. This problem can be solved by observing the path coefficient in the SEM model.

Through this paper, several contributions are made. First, from the perspective of topic selection, this paper explores a variety of internal and external factors that affect the quality and level of EAID, providing help for corporations to improve their quality and level of relevant disclosure. Second, from the perspective of methodology, this paper uses the structural equation model to explore the relationship between external pressure, corporate performance, corporate governance and EAID for the first time, which is a methodological innovation. Third, for each of the latent variables in this paper, four observed variables (supported by previous literature) are allocated to analyze not only the external relationship between the observed variables, but also the path coefficient between observed variables and latent variables. In other words, we can see the degree of contribution 'observed variables' make in terms of latent variables (such as how much the state-owned equity ratio contributes to the external pressure), which cannot be observed by the traditional regression analysis method previously used.

The remainder of the article is organized as follows. Section 2 puts forward the corresponding hypothesis, based on four mature theories. Section 3 describes the sample selection and constructs variables. Section 4 builds the SEM model and explains the results to verify the hypotheses. Variance inflation coefficient and regression analysis are then used to test the collinearity of variables and robustness of the conclusion, respectively. Section 5 draws the conclusion and the enlightenment offered by this paper. Based on the above,

Section 6 summarizes the shortcomings of the research and raises possible directions for future research.

## 2. Theoretical Analysis and Research Hypothesis

This section is divided into four parts. In the first three parts, based on previous literature, we narrate the relationship between the above-mentioned three parts (i.e., external pressure, corporate performance, corporate governance) and environmental accounting information disclosure.

### 2.1. Reputation Dynamic Model

Formbrun and Shanley's paper (1990) [21] in the Academy of Management Journal is generally considered to be the establishment of the "Corporate dynamic reputation model". According to this model, for a mature corporation, economic performance is not the sole criterion to judge its success. The current and past behaviors of a corporation send signals to the external public in different ways to influence the reputation of the corporation, which, in turn, affects the future actions of the corporation, forming a "cycle", as illustrated in Figure 1 below.

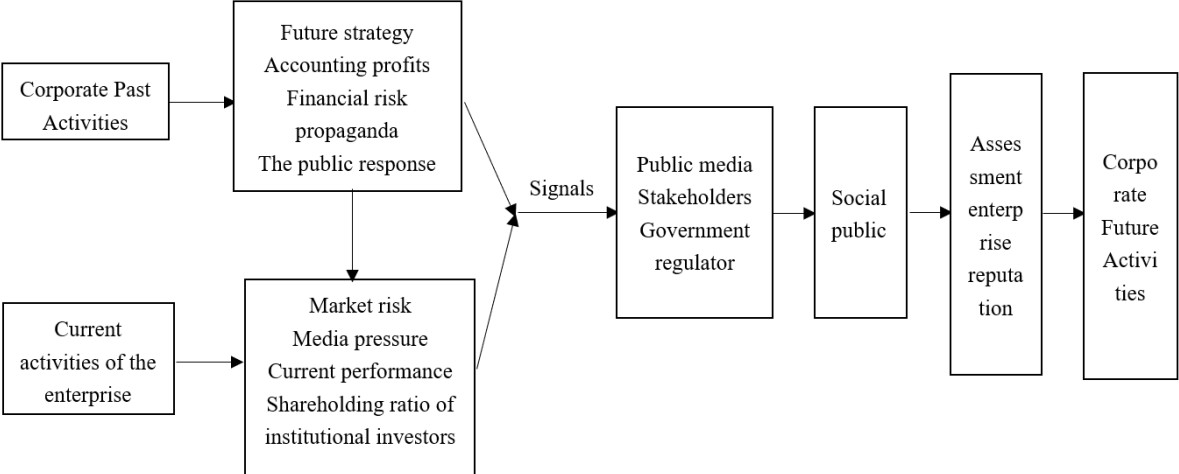

**Figure 1.** Corporate Dynamic Reputation Model.

From the model of reputation formation, we learn that after public media and government regulatory agencies receive the signals formed by various current and past activities of the corporation, they transmit the signals to the public to exert external pressure on the corporation's reputation, thus, stimulating the corporation's future activities and behaviors. According to the "rational economic assumption", the basic assumption of western economics, companies all want to enter a virtuous circle to make their futures better. So, based on this, we propose the following hypothesis:

**Hypothesis 1 (H1).** *External pressure has significantly positively affected the environmental accounting information disclosure of listed companies.*

### 2.2. Legitimacy Theory

Derived from political economy, legitimacy theory mainly concerns the positioning of corporations in a social system. It believes that society, politics and economy are indivisible. Without the whole system of political, social and institutional frameworks, the study of economic problems may become "empty talk". Since corporations exist in such a system, the theory of legitimacy believes that corporations are connected with certain social contracts. Only if they abide by the social contract, and the public recognizes their legitimacy, can they achieve sustainable development. Corporations' information disclosure is considered

to be an important means for the management to influence the external perception of the corporation [22]. Consequently, ceteris paribus, the social legitimacy of corporations largely depends on the information disclosure they make.

According to the legitimacy theory, an important purpose for corporations in disclosing environmental accounting information is maintenance of their own "legitimacy", so that the decision-making behaviors of corporations are consistent with the moral standards that the public suppose should apply. Since the social legitimacy of corporations are monitored by the field of public policy, rather than the capital market, environmental accounting information disclosure should be more associated with public external pressure [23]. Based on this, we propose three sub-hypotheses on the basis of Hypothesis H1:

**Hypothesis 1a (H1a).** *There is a significant positive correlation between the intensity of government supervision and the quality of corporate environmental accounting information disclosure.*

**Hypothesis 1b (H1b).** *The public opinion of social media promotes the disclosure of corporations' environmental accounting information.*

**Hypothesis 1c (H1c).** *The external pressure of loans from banks and other financial institutions on corporations positively promote the disclosure of environmental accounting information of corporations.*

### 2.3. Signal Transmission Theory

The signal transmission theory was proposed by Michael Spence in 1973 [24]. According to this theory, under the circumstance of information asymmetry, the party with the information advantage in the market needs to transmit information to the party with the information disadvantage, so as to inform the real level and quality and make the transaction more efficient.

Combined with the reputation theory above, and applied to the disclosure of environmental accounting information, corporations with good operating conditions are more willing to disclose environmental accounting information in order to send a positive signal to the outside world, gain a good reputation and attract more investors. The mechanism of the above-mentioned process is illustrated in Figure 2.

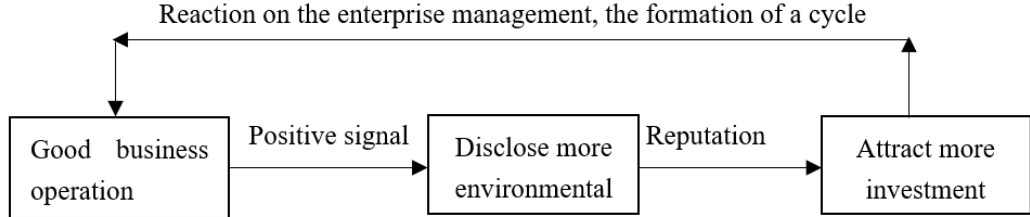

**Figure 2.** Mechanism of Signaling Model.

Based on this, we propose the following hypothesis:

**Hypothesis 2 (H2).** *Corporations with good performance disclose a wider range of environmental accounting information with higher quality.*

### 2.4. Voluntary Information Disclosure Theory

Voluntary disclosure refers to the disclosure of information that can be flexibly adjusted according to one's own wishes and needs in addition to mandatory disclosure [25]. Compared with western developed countries, the development of environmental accounting in China was relatively slow. On account of relevant laws and regulations not being sound, the government could not systematically require corporations to disclose environmental accounting information. Therefore, most environmental accounting information belongs to the voluntary information disclosure part of disclosure.

For corporations, voluntary information disclosure is not only conducive to reducing agency costs, enhancing supervision over management and improving information transparency, but is also beneficial to investor relationship management and enhancing corporate values recognized by the market [26]. Nevertheless, at present, in the process of environmental accounting information disclosure, there is a very interesting phenomenon called "beating faster cattle", which means, corporations that disclose more relevant information have greater possibility of being punished because of an unnoticed information violation (similar to "He who has done more made more mistakes"). In this case, corporate governance behavior is particularly important. Previous literature has shown that more disclosure of social responsibility information improves the social reputation of corporations [27,28]. Good corporate governance also emboldens corporations to submit more voluntary information disclosure. Even if the disclosure reveals some problems, stable management is able to deal with them and remedy them. Based on this, we propose the following hypothesis:

**Hypothesis 3 (H3).** *Corporate governance is positively correlated with corporate environmental accounting information disclosure.*

## 3. Materials and Methods

### 3.1. Sample Selection

This paper selected 200 listed companies in China's A-share market from 2015 to 2019 as research samples.

Reason for time selection: China adopted the newly revised Environmental Protection Law at the eighth session of the Standing Committee of the 12th National People's Congress, on 24 April 2014, and officially implemented it on 1 January 2015. The law not only improves the basic system of China's environmental protection, but also strengthens social responsibility of corporations (especially the responsibility of environmental protection and governance), and increases the intensity of punishment for illegal acts (such as illegal pollution discharge and environmental damage), which has greatly encouraged the self-supervision of corporations. Although existing research suggests that regulatory law can generate extra costs and risks for firms, it has a positive effect on the voluntary disclosure of corporations [29,30]. After the implementation of the Law in 2015, corporations disclosed environmental accounting information in a more efficient and normative way.

Reasons for 200 samples: As there are no special laws and regulations on environmental accounting information in China, listed companies have great discretion regarding information disclosure, which causes some inconsistency in information disclosure. Therefore, in order to ensure the comparability and integrity of sample data, we tried to select companies having the 13 kinds of information which could be integrally obtained in CSMAR for 13 kinds of observation variables of environmental accounting information disclosure. To achieve this goal, in the "Environmental Research" section of CSMAR, we selected the "Resource consumption list of listed companies" with the least available data as the "basic table", and obtained data of 113 listed companies. Then, the intersection of the 12 other kinds of information was selected to obtain the remaining 87 data. Meanwhile, due to the huge impact of COVID-19 on all industries from 2020 to the present, the time span of our sample did not include the two recent years.

The sample companies were involved in eight industries, including energy, transportation and the electronics industry, etc. In the process of sample selection, they were screened as follows:

I. Samples with incomplete data for 3 years were excluded;
II. Samples with negative net worth were excluded;
III. In order to avoid abnormal extreme values, ST company samples were excluded;
IV. The business model, reporting structure and major accounting items of the financial industry are different from those of other industries, so we excluded financial corporation samples.

After the above screening criteria screening were applied and the data obtained winsorized, we obtained a total of 200 company samples and 5000 valid observations (5000 valid observations = 200 samples $\times$ 12 environmental subjects for measuring environmental accounting information + 13 observation variables for latent variables).

Environmental accounting information data and explanatory variable data (external pressure, corporate performance and corporate governance) of the sample corporations were mostly obtained from the "environmental research" section, "financial statements" and "financial index analysis" of CSMAR, respectively. As for a small number of missing values in the database, we sorted them manually through the corresponding corporate annual report, sustainable development report and authoritative media reports. Among the external pressures, the "Janis–Fadner coefficient", which measures the pressure of media public opinion, came from the "public opinion monitoring index plate" of Wind Database. The proportion of state-owned shares came from "shareholder research plate" in Eastern Wealth Net. All data in this paper were available through public channels.

### 3.2. Variable Definitions
### 3.2.1. Explained Variables

The explained variable in this paper was the disclosure of environmental accounting information. In order to increase the objectivity and persuasion of the conclusion, we set two observation variables for the classification and scoring of environmental accounting information, respectively, which were the following: The level of EAID (measured by Rankins CSR Ratings Global MCTI Social Responsibility Reporting Rating System 2012 edition (C11–C14 in the environment section)) and the quality of EAID (measured by the environmental project scoring method commonly used in the current academic community). Quality meant to what extent the content of the information included in EAID was specifically quantified, and level inspected how many items were included in the EAID. In a word, quality indicated the depth of a corporate's EAID while level showed the width of EAID. Rankins CSR Ratings Global MCTI Social Responsibility Reporting and Rating System 2012 (hereinafter referred to as "MCTI System") is an authoritative indicator system, reflecting the CSR performance of listed companies in Asian capital markets. It quantifies CSR into specific indicators and provides an objective tool for assessing the CSR performance. The MCTI system principally includes six aspects: economic performance, labor and human rights, environment, fair operation, consumer and community participation and development, which focus on the disclosure scope of environmental accounting information. This paper utilized the "environment" part of the system, which divides the environmental accounting information disclosed by corporations into four categories. Since the system mainly targets the scope of environmental accounting information disclosure and rarely involves precise values, we only set "Mentioned"—1 point and "not mentioned"—0 point, in terms of assigning criteria, as shown in Table 1 below.

**Table 1.** MCTI Social Responsibility Report Rating System 2012 environmental section.

| Information Classification | Examples of Containing Information | Scoring Standards |
|---|---|---|
| Overall environment management information | Annual Environmental Investment | The scoring standards of corporate disclosure level are as follows: Mention: 1 point Not mentioned: 0 point Cumulative score of corporate disclosure level = Cumulative sum of the four projects |
| Pollution prevention information | Identification and measurement of discharge pollution and waste, and measures to control pollution sources | |
| Information on sustainable resource use | Measure, record and report water usage | |
| Information on climate change mitigation and adaptation | Greenhouse gas reduction measures | |

For quality of environmental accounting information disclosure, we evaluated it through environmental item scoring method. This method was firstly put forward by

Wiseman (1982) [31], and classifies environmental accounting information disclosed by corporations into the following four categories:

(I).   Projects directly related to economic factors;
(II).   Projects related to environmental litigation;
(III).  Pollution reduction projects;
(IV).  Other environment-related projects that do not fall into any of the above categories.

Among these four categories, there are 18 sub-items, each of which is assigned with a score based on the level of detail disclosed. Cormier and Magnan (2003) [32], Al-Tuwaijri et al. (2004) [8] and Clarkson (2008) [10] also used this scoring method in their related studies. As China has not yet established a system of environmental accounting disclosure, most of the information possesses the character of voluntary disclosure of environmental accounting information. Under such circumstances, the method and content of disclosure may vary a lot between different corporations. In such a condition, we set "Measures for environmental accounting information disclosure of corporations and institutions", an officially passed document by the Ministry of Environmental Protection in a ministerial conference in 2014, as a benchmark, combined with the newly-advised "Environment Protection Law" and environmental item scoring method designed by Wiseman (1982) [31] to divide environmental accounting information disclosed by corporations into the following three categories and nine specific items. The details and scoring rules are in Table 2.

**Table 2.** Corporate Environmental accounting information Disclosure Score Index System.

| Category | Disclosure of Project Contents | Supposed the Standard |
| --- | --- | --- |
| Environmental Management Information | Resource consumption information | The scoring standards of corporate disclosure are as follows: Quantitative, precisely monetized information—3 points; Qualitative, specific non-monetized information—2 points; General mention of information—1 point; Not mentioned—0 point. Corporate environmental accounting information disclosure score = cumulative score of nine items. |
| | ISO environmental system certification information | |
| | Emergency environmental accidents, environmental illegal incidents, environmental petition cases | |
| Environmental performance information | Measures to improve (or remedy) the ecological environment | |
| | Pollutant discharge and emission reduction treatment | |
| | Environmental investment, loans and related technology research and development | |
| Environmental responsibility information | Implementation of cleaner production | |
| | Environmental protection objectives, concepts and management systems | |
| | Environmental emergency response mechanism | |

### 3.2.2. Explanatory Variables

Through reorganizing previous literature, this paper selected four observation variables to specifically observe the potential variables of external pressure: bank loans, proportion of state-owned shares, whether audited by the "Big Four" major auditing institutions and the pressure index of public media opinion. Among these, bank loans represented the pressure imposed by creditors on corporations. The higher the proportion of bank loans a corporation has, the more attention banks pay to the corporation's operating conditions and solvency. When the loan exceeds a certain amount, the bank imposes corresponding restrictions on the use of loans. Hence, bank loans place certain pressures on corporations. As an organic part of national finance, state-owned corporations usually need to undertake more complex social responsibilities. This is the reason why corporations with a high proportion of state-owned shares are expected to "disclose more social responsibility information". Compared with general accounting firms, choice of the international "Big Four" accounting firms not only reflects their own good operation and business standards to the outside world, but also increases the external pressure faced by corporations, because the

"Big Four" accounting firms face higher litigation risks than other general accounting firms and, therefore, pay more attention to the disclosure of non-financial information, such as corporate social responsibility, which includes EAID. According to reputation theory, when public opinion focuses on environmental accounting information disclosure of corporations, corporations with good operating conditions tend to increase related disclosure to "defend" themselves and "save" their reputation. In this regard, we introduced the Janis-Fadner coefficient (J-F coefficient for short), which is commonly used in international academic circles to quantify the pressure of social public opinion. The J-F coefficient is an index jointly proposed by scholars Janis and Fadner to measure the pressure caused by public opinion supervision on corporations [33]. Its calculation formula is shown in Formula (1), as follows:

$$
\text{J} - \text{F} = \begin{cases} \frac{(e^2 - ec)}{t^2}, & \text{if } e > c \\ 0, & \text{if } e = c \\ \frac{(ec - c^2)}{t^2}, & \text{if } e < c \end{cases} \tag{1}
$$

In the above Formula (1), "e" represents the number of positive corporate news reported by media, and "c" represents the number of negative corporate news, t = e + c. The value range of the J-F coefficient is $(-1, 1)$. From the formula, we know that the larger the "e" value is, the more the coefficient tends to 1, indicating that the public opinion pressure borne by the corporate is less. On the contrary, the higher the value of "c" is, the more the coefficient tends to $-1$, indicating that the corporate bears more pressure from public opinion. As a result, the J-F coefficient is inversely proportional to the pressure of public opinion borne by corporations.

In the process of exploring the relationship between corporate performance and EAID, we selected earnings per share, net profit growth rate, total asset turnover and financial leverage as the four observation variables to respectively represent the overall operating results of the corporation, the company's profitability growth ability, sales ability and the utilization efficiency of debt financing. Among these, earnings per share referred to the current net profit of common shareholders divided by the weighted average of outstanding common shares, so as to calculate earnings per share, which is the basic and core index to evaluate the profitability of listed companies. In addition to profitability, while evaluating the performance of a corporation, we also needed to observe its capital utilization efficiency. For a corporation, its capital sources mainly come from two aspects: internal capital and external debt. Therefore, the net profit growth rate (NPR), which reflects the efficiency of the company using its own capital, and the financial leverage (DFL), which reflects the efficiency of the company using debt financing, can comprehensively measure the ability of the company to use its capital. Sales revenue is the basis for most corporations to obtain profits and develop, and sales capacity largely determines the future development space of a corporation. So, we used total asset turnover (TAT) to measure the corporation's sales capacity. To sum up, these four parts basically reflect the performance of a corporation.

On account of its abstract coverage, corporate governance itself cannot be quantified precisely, so this paper chose four indicators to measure it: ownership concentration, proportion of independent directors, management shareholding and equity balance. Ownership concentration is not only the main index to measure the degree of ownership concentration of a company, but also to measure the stability of the company. Literature has shown that there is an inverted "U-shape" relationship between ownership concentration and corporate performance. Moderate ownership concentration is more conducive to corporate governance mechanism playing a role and maximizing corporate governance efficiency [34]. The degree of equity checks and balances refers to the controlling degree of the company's top shareholders compared with the largest shareholder. The higher the degree of equity balance is, the stronger the controlling degree of external shareholders is (compared with controlling shareholders). Under such a consequence, external shareholders have stronger supervision motivation and ability. To some extent, a moderate number of external shareholders prevents controlling shareholders infringing on the rights and interests of

minority shareholders. Therefore, the appropriate degree of ownership balance optimizes the ownership structure of corporations, and improves corporate performance. However, an excessive degree of equity balance may also lead to equity dispersion, which has a significant negative impact on corporate performance [35]. Thus, both ownership concentration and ownership balance are important indicators to reflect the characteristics of corporate equity structure, and can also be used as observation variables to explore corporate governance. Independent directors can effectively reduce insider control, and have the right to make their own independent judgment when major problems occur in the corporation, so as to protect the rights and interests of minority shareholders to the greatest extent. So, increase in the proportion of independent directors hired by corporations is helpful to enhance the objectivity and independence of the board of directors, and, thus, limit harmful behaviors to shareholders to a certain degree. Smith et al. (1990) [36] showed that improvement of corporate performance was partly attributed to the ownership of certain equity by the management, which increased the motivation of managers to create wealth. As a result, management ownership directly affects corporate governance.

*3.3. Table of Observation Variable Definitions*

Combined with previous literature, this paper selected 12 observation variables to observe three latent variables, namely external pressure, corporate performance and corporate governance, which were difficult to directly observe. All indices used for observation variables were supported by former literature. The definition table of observed variables is given in Table 3.

**Table 3.** Definition table of observed variables.

| Variable Types | Latent Variables | Observation Variable | Variable Definitions | Literature Sources |
|---|---|---|---|---|
| Explanatory variables | External pressure | Loans (Loans) | The sum of the company's short-term and long-term borrowings; | [37] |
| | | Proportion of state-owned Shares (SOE) | Total number of state-owned shares/corporations | [1] |
| | | Whether audited by Big Four accounting firm (Big4) | Dummy variables; Assigned 1 if they are, 0 if they are not; | [38] |
| | | Social Public Opinion Pressure Index (J-F) | Janis-fadner coefficient (J-F coefficient) | [4] |
| | Corporate performance | Earnings per share (EPS) | Net profit/Total number of shares issued; | [39] |
| | | Profitability (NPR) | Net profit growth rate = (Current year net profit—last year net profit)/Last year net profit | [5] |
| | | Operating Capacity (TAT) | Total asset turnover = sales revenue/Total assets | [40] |
| | | Financial Leverage (DFL) | Change in earnings per common share/Change in EBIT | [4] |
| | Corporate governance | Ownership Concentration (Herf5) | The sum of squares of the shareholding ratio of the top five shareholders; | [41] |
| | | Equity Balance degree (ERR) | Shareholding ratio of the 2nd–5th largest shareholder/Shareholding of the largest shareholder; | [42] |
| | | Management Shareholding (MH) | Management shareholding/Total shares; | [43] |
| | | Proportion of independent directors (RID) | (Proportion of independent directors) = Number of independent directors/Total number of directors | [44] |

| Variable Types | Latent Variables | Observation Variable | Variable Definitions | Literature Sources |
|---|---|---|---|---|
| Explained variable | Corporate environmental accounting information disclosure | Management Shareholding (MH) | Management shareholding/Total shares | [1,7,10,31,42–44] |
| | | Equity Balance degree (ERR) | Shareholding ratio of the 2nd–5th largest shareholder/Shareholding of the largest shareholder | |
| | | Level of Environmental accounting information disclosure (EDI) | Rankins CSR Ratings Global MCTI Social Responsibility report Rating System 2012 edition | |
| | | Quality of Environmental accounting information disclosure (EDQ) | Environmental Scoring Guidelines | |
| Control variables | Corporate Size (Size) | The natural log of total assets | Ln (Total assets) | |
| | Industry Type (Industry) Degree of Debt | The bureau of Statistics publishes industry classifications Asset-liability ratio | Total liabilities/total assets | |

## 4. Results

*4.1. Descriptive Statistics*

Descriptive Statistics of Initial Data

Tables 4 and 5 below show the corresponding statistics of quality and level of environmental accounting information disclosure in the 200 sample companies for the period 2015–2019. Statistics of observation variables are shown in Table 6. The sample corporations came from eight industries: Energy industry, Transportation, Life service industry, Electronics industry, Food industry, Mechanical and Technology industry, Real estate/Construction industry and Pharmaceutics industry.

**Table 4.** Descriptive Statistics of EAID's Quality (by Industry).

| Industry | Sample | Mean | Standard Deviation | Median | Minimum | Maximum |
|---|---|---|---|---|---|---|
| Energy industry | 62 | 210.84 | 13.30 | 178 | 1 | 730 |
| Machinery/technology industry | 69 | 108.97 | 10.77 | 83 | 1 | 436 |
| Electronics industry | 9 | 152.67 | 15.84 | 106 | 7 | 497 |
| Real estate/construction industry | 12 | 117.91 | 11.31 | 131 | 13 | 269 |
| Life service industry | 10 | 102.3 | 10.49 | 100.5 | 1 | 258 |
| Food industry | 11 | 141.8 | 10.25 | 123 | 85 | 234 |
| Pharmaceutical industry | 18 | 124.1 | 11.4 | 100 | 25 | 325 |
| Transportation | 9 | 150.44 | 7.14 | 157 | 51 | 233 |
| Overall average | - | 138.63 | 11.31 | 122.31 | 23 | 372.75 |

**Table 5.** Descriptive Statistics of EAID's Level (by Industry).

| Industry | Sample | Mean | Standard Deviation | Median | Minimum | Maximum |
|---|---|---|---|---|---|---|
| Energy industry | 62 | 79.13 | 5.83 | 78.5 | 3 | 232 |
| Machinery/technology industry | 69 | 44.16 | 11.04 | 37 | 4 | 124 |
| Electronics industry | 9 | 57.67 | 4.47 | 54 | 7 | 150 |
| Real estate/construction industry | 12 | 61.33 | 6.08 | 57.5 | 5 | 153 |
| Life service industry | 10 | 51.7 | 9.90 | 45 | 1 | 125 |
| Food industry | 11 | 59.63 | 8.60 | 60 | 22 | 109 |
| Pharmaceutical industry | 18 | 72.67 | 5.48 | 56 | 15 | 296 |
| Transportation | 9 | 79.11 | 8.77 | 56 | 20 | 152 |
| Overall average | 200 | 63.17 | 7.52 | 55.5 | 9.63 | 167 |

**Table 6.** Descriptive Statistics of Observation Variables.

| Variable | N | Minimum | Maximum | Mean | Standard Deviation |
|---|---|---|---|---|---|
| Loans | 200 | 0 | 132,868,500,000 | 4,973,085,898.6 | 12,976,193,546.43 |
| J-F coefficient | 200 | 0.218 | 0.945 | 0.658 | 0.124 |
| Proportion of state-owned equity (SOE) | 200 | 0 | 0.932 | 0.229 | 0.268 |
| Whether audited by "Big Four" ("Big Four") | 200 | 0 | 5 | 4.76 | 0.791 |
| Net Profit Growth Rate (NPR) | 200 | −16.287 | 378.07 | 2.639 | 27.146 |
| Earnings per share (EPS) | 200 | −1.03 | 2.190 | 0.401 | 0.432 |
| Debt financial leverage (DFL) | 200 | −9.297 | 155.698 | 2.68 | 11.28 |
| Total asset turnover (TAT) | 200 | 0.124 | 2.073 | 0.593 | 0.339 |
| Ratio of independent director (RID) | 200 | 0.038 | 3.03 | 0.730 | 0.605 |
| Management shareholding (MH) | 200 | 0.136 | 0.663 | 0.373 | 0.052 |
| Ownership concentration (Herf5) | 200 | 0 | 60.90 | 8.541 | 15.229 |
| Disclosure quality score (Quality Score) | 200 | 0.003 | 0.718 | 0.192 | 0.132 |
| Disclosure level score (Level Score) | 200 | 1 | 730 | 148.03 | 118.757 |

*4.2. Correlation Analysis*

To verify the hypothesis preliminarily, we first analyzed the correlation between explanatory variables, the quality and level of environmental accounting information disclosure and the explained variables (12 observational variables), by using SPSS.26. The correlation analysis matrix is shown in Table 7.

As for public pressure, the above table shows that sample companies' loans, media opinion orientation (J-F) and proportion of state-owned shares (SOE) were significantly positively correlated with their environmental accounting disclosure quality and level. This showed that pressure from banks, government regulators and public media encouraged listed companies to increase information disclosure of environmental accounting, which indicated that external pressure from government and media positively enhanced corporation's EAID [45]. In particular, the proportion of state-owned equity (SOE) in the disclosure quality and level were of high correlation, which strongly suggested that, for Chinese state-owned corporations, having relatively low risk, one of the most important goals was to realize social benefit maximization [46], and the implementation of social responsibility was no longer "the icing on the cake" (merely ornament for financial report), but a political objective [42]. To make a "CSR-responsible" example for others, state-owned enterprises might focus more on affairs related to EAID, such as green innovation, etc. [47]. Therefore, the higher the proportion of state-owned shareholders, the more obvious the political targets pursuit by those corporations. However, the results did not show a strong correlation with whether they were audited by the Big 4 accounting firms. Chances are that complete laws and regulations have not been established in China's EAID. Therefore, taking cost into consideration, even top accounting firms, such as the "Big Four", only have limited time and focus on whether the audited unit has any illegal fraud, rather than the fragmented environmental accounting information which often hidden in the notes to the

company's financial statements. In general, external pressure had a significant positive impact on the environmental accounting information disclosed by corporations, which was consistent with the views of previous literature [1,7].

**Table 7.** Correlation Matrix.

| | Loans | J-F | SOE | Big4 | NPR | EPS | DFL | TAT | ERR | RID | MH | Herf5 | EDQ | EDI |
|---|---|---|---|---|---|---|---|---|---|---|---|---|---|---|
| Loans | 1.000 | 0.243 *** | 0.415 *** | 0.041 | −0.006 | 0.031 | −0.013 | −0.056 | −0.033 | 0.132 ** | −0.185 *** | 0.344 *** | 0.179 *** | 0.235 *** |
| J-F | 0.243 *** | 1.000 | 0.157 | −0.084 | 0.108 | 0.089 | −0.110 | 0.114 | −0.076 | 0.081 | −0.080 | 0.164 | 0.163 | 0.204 |
| SOE | 0.415 *** | 0.154 ** | 1.000 | 0.032 | 0.097 | −0.003 | 0.078 | −0.032 | −0.216 | 0.147 | −0.406 | 0.439 | 0.247 | 0.308 |
| Big4 | 0.041 | −0.083 | 0.031 | 1.000 | 0.024 | −0.026 | 0.022 | 0.041 | −0.091 | 0.195 | −0.124 | 0.066 | 0.097 | 0.086 |
| NPR | −0.006 | 0.108 | 0.097 | 0.024 | 1.000 | 0.021 | −0.009 | −0.013 | 0.014 | −0.032 | −0.024 | 0.014 | 0.137 | 0.046 |
| EPS | 0.031 | 0.087 | −0.002 | −0.026 | 0.021 | 1.000 | −0.200 | 0.123 | 0.066 | 0.021 | 0.040 | 0.221 | 0.139 | 0.238 |
| DFL | −0.013 | −0.110 * | 0.078 | 0.022 | −0.009 | −0.2 *** | 1.000 | −0.017 | −0.040 | −0.037 | −0.060 | −0.085 | −0.041 | −0.031 |
| TAT | −0.056 | 0.118 * | −0.037 | 0.042 | −0.013 | 0.121 ** | −0.018 | 1.000 | −0.102 | −0.015 | 0.005 | 0.187 | 0.236 | 0.150 |
| ERR | −0.033 | −0.077 | −0.215 *** | −0.091 | 0.014 | 0.067 | −0.040 | −0.102 * | 1.000 | −0.109 | 0.310 | −0.510 | −0.140 | −0.162 |
| RID | 0.132 ** | 0.085 | 0.142 ** | 0.196 *** | −0.032 | 0.018 | −0.037 | −0.009 | −0.110 ** | 1.000 | −0.111 | 0.218 | 0.070 | 0.165 |
| MH | −0.185 *** | −0.077 | −0.408 *** | −0.123 ** | −0.024 | 0.038 | −0.060 | 0.008 | 0.310 *** | −0.107 ** | 1.000 | −0.273 | −0.228 | −0.328 |
| Herf5 | 0.344 *** | 0.163 ** | 0.438 *** | 0.066 | 0.014 | 0.221 *** | −0.085 | 0.186 *** | −0.510 *** | 0.217 *** | −0.273 *** | 1.000 | 0.216 | 0.299 |
| EDQ | 0.179 *** | 0.161 ** | 0.248 *** | 0.096 * | 0.137 | 0.140 ** | −0.040 | 0.233 *** | −0.139 ** | 0.068 | −0.228 *** | 0.216 *** | 1.000 | 0.744 |
| EDI | 0.235 *** | 0.199 *** | 0.311 *** | 0.084 | 0.046 | 0.239 *** | −0.030 | 0.144 ** | −0.161 ** | 0.159 ** | −0.330 *** | 0.299 *** | 0.744 *** | 1.000 |

Note: ***, ** and * indicate statistically significant results at 1%, 5% and 10% levels, respectively.

For corporate performance, earnings per share (EPS) and total asset turnover (TAT), which reflect the profit level of common stock of listed companies, were significantly positively correlated with the quality and level of EAID. At the same time, although there was no significant correlation between asset profit margin and disclosure quality, there was a positive correlation between asset profit margin and disclosure level of 0.05 significance level, which indicated that corporate profitability positively affected environmental accounting information disclosure, whether in terms of quality or level [48]. However, (DFL) was inversely proportional to environmental accounting information disclosure, which indicated that, compared with key indicators, such as net profit growth rate and earnings per share, investors were keen to pay attention to financial leverage. Listed companies with high financial risks tend have lower disclosure of environmental accounting information, which seem to be "minor details". On the contrary, considering that "the tongue cuts the throat", such a company might deliberately conceal environmental accounting information which is detrimental to itself. To sum up, corporate performance also had a positive role in promoting its EAID [49].

In terms of corporate governance, the above table shows that ERR was significantly negatively correlated with environmental accounting information disclosure. However, there was a significant positive correlation between ownership concentration (Herf5) and disclosure. It is generally believed that equity balance degree and equity concentration degree are "opposite indicators". The greater the degree of equity balance is, the more dispersed the company's equity is. Meanwhile, the discourse power of the largest shareholder is weaker than that of the second to the Nth shareholder as well. The degree of ownership concentration reflects the concentration of major shareholders. It is generally

believed that a higher degree of equity checks and balances is conducive to democratic decision-making and effectively inhibits the infringement of the interests of minority shareholders by the largest shareholder; thus, facilitating the disclosure of information related to social responsibility. In contrast, in a company with concentrated equity it is easy for the largest and the majority shareholders to form a "small group" to totally control the whole corporation; thus, eroding the interests of minority shareholders. The results in the table above support the following hypothesis. Since large shareholders have higher shares and are able to gain more benefits from the improvement of corporate performance than minority shareholders, they are highly connected to the corporation, and, thus, are more eager to improve all aspects of corporate performance ("all aspects" including "social image"). To achieve this, they are willing to pay more attention to reflect corporate social responsibility (CSR) by disclosing more environmental accounting information. Of course, more information disclosure means more disclosure costs. Although the controlling shareholder may have some negative effect on accounting conservatism [30], high ownership concentration means high decision-making efficiency of the company [46]. Therefore, measures that are conducive to EAID but increase the company's explicit cost, proceed successfully. In contrast, for a company with decentralized equity, the decision-making is more democratic but less efficient, and measures that significantly increase explicit costs of the company are subject to great resistance from minority shareholders. Moreover, the dispersion of holdings means that shareholders' earnings are diluted, and shareholders are less enthusiastic about governance. As a result, companies with concentrated equity disclose more environmental accounting information. With this in mind, it is not hard to understand why there was significant negative correlation between management shareholding and EAID. Most listed companies have certain incentive plans. Although these plans vary in different companies, they are often linked to company performance indicators (usually high-profile targets which concern shareholders and investors the most, such as net profit). In other words, in companies implementing equity incentive plans, net profit affects management compensation. While trying to generate revenue for the company, management cut all sorts of seemingly "unnecessary" costs, such as the costs associated with disclosure of environmental accounting information.

To sum up, based on the correlation matrix between observed variables and explained variables, we conducted a preliminary view on the relationship between observed variables subordinate to the three latent variables and environmental accounting information disclosure.

*4.3. SEM Model and Analysis*

In this part, we further explore the relationship between external pressure, corporate performance, corporate governance and corporate environmental accounting information disclosure by constructing a structural equation model. We applied AMOS.26.0 to calculate the model, then Maximum likelihood method was used to estimate parameters. In order to eliminate dimensional differences, all the variables, except "loans" and "the "Big Four" auditing", were standardized. As the observed values of "loans" and "whether there were Big Four major audits" were significantly different from other items after standardized treatment, we normalized these two items.

The brief path diagram of the structural equation model is illustrated in Figure 3.

Then we added observation variables, explanatory variables and residual terms into the preliminary model to get the integral model. After calculations, the structural equation model diagram and path coefficient table were obtained, as is shown in Figure 4.

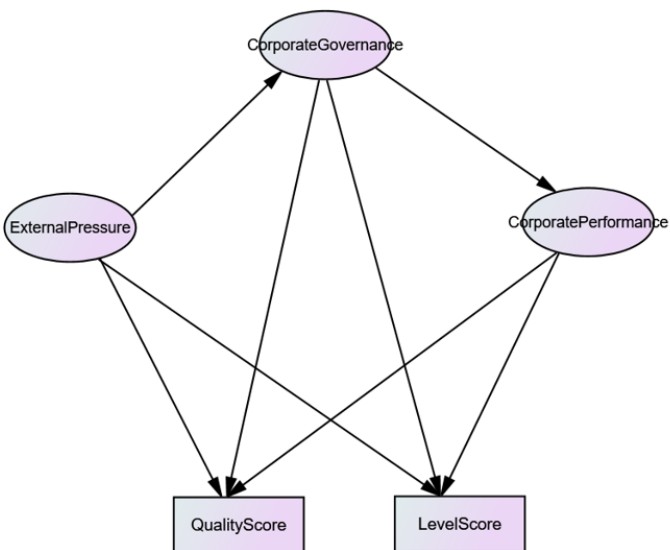

**Figure 3.** Model Brief Path Diagram.

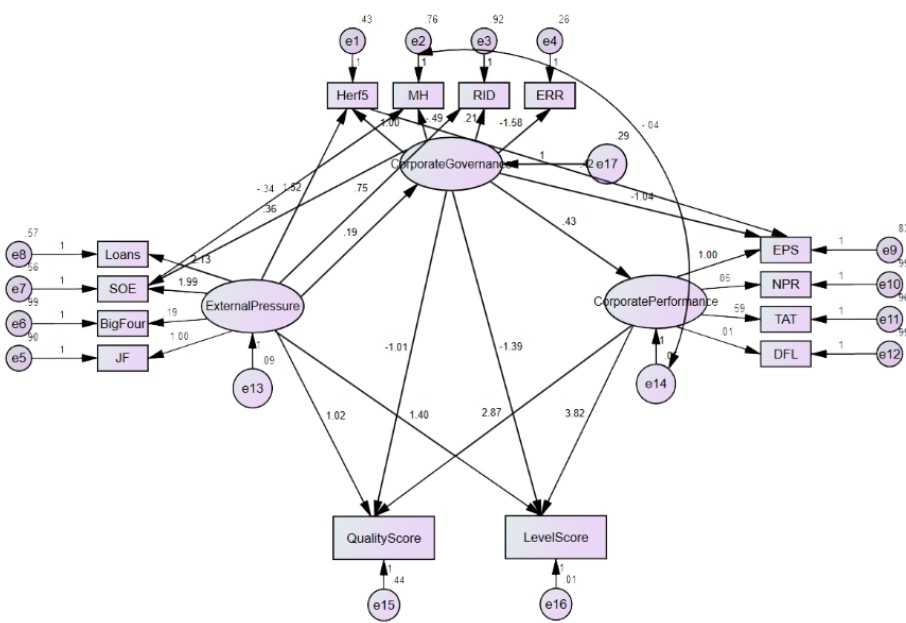

**Figure 4.** Structural Equation Model.

From the above chart we could see that, when using the maximum likelihood method to estimate the integral model, the factor loading value of the three paths were set to be a fixed parameter "1":

①    External pressure → J-F coefficient;
②    Business performance → Earnings per share;
③    Corporate governance → Ownership concentration.

In the SEM, because there is no virtual unit for latent variables, in order to compare these factors' loading value of paths, for each latent variable having 4 paths, we needed to select one path and set its factor loading value as "1" to be the "reference indicator", so as to compare it with others as a benchmark. As is shown in Table 8, the "reference indicator" did not need a significant coefficient test. This is a good method for managing the model of DF (degree of freedom).

**Table 8.** Path Coefficients of Structural Equation Model.

| | Estimate | S.E. | C.R. | *p* Value |
|---|---|---|---|---|
| External Pressure → SOE | 1.986 | 0.600 | 3.310 | *** |
| External Pressure → "Big Four" | 0.193 | 0.282 | 0.683 | 0.495 |
| External Pressure → Loans | 2.131 | 0.630 | 3.382 | *** |
| External Pressure → J-F coefficient | 1.000 | - | - | - |
| Corporate Performance → DFL | 0.008 | 0.231 | 0.034 | 0.973 |
| Corporate Performance → NPR | 0.054 | 0.232 | 0.231 | 0.817 |
| Corporate Performance → TAT | 0.593 | 0.361 | 1.641 | 0.101 |
| Corporate Performance → EPS | 1.000 | - | - | - |
| Corporate governance → ERR | −1.585 | 0.457 | −3.471 | *** |
| Corporate governance → MH | −0.487 | 0.150 | −3.246 | 0.001 |
| Corporate governance → RID | 0.214 | 0.148 | 1.440 | 0.150 |
| Corporate governance → Herf5 | 1 | − | − | − |
| External Pressure → Quality Score | 1.016 | 0.387 | 2.624 | 0.009 |
| External Pressure → Level Score | 1.396 | 0.457 | 3.055 | 0.002 |
| Corporate Performance → Quality Score | 2.868 | 0.882 | 3.253 | 0.001 |
| Corporate Performance → Level Score | 3.818 | 1.348 | 2.833 | 0.005 |
| Corporate governance → Quality Score | −1.006 | 0.986 | −1.020 | 0.308 |
| Corporate governance → Level Score | −1.392 | 1.339 | −1.039 | 0.299 |
| External Pressure → Corporate governance | 0.195 | 0.217 | 0.895 | 0.371 |
| External Pressure → Herf5 | 1.520 | 0.484 | 3.142 | 0.002 |
| External Pressure → RID | 0.214 | 0.148 | 1.440 | 0.150 |
| Corporate governance → SOE | 0.359 | 0.141 | 2.541 | 0.011 |
| Corporate governance → Corporate Performance | 0.434 | 0.368 | 1.179 | 0.238 |
| Herf5 → EPS | 0.418 | 0.118 | 3.538 | *** |
| SOE → MH | −0.338 | 0.067 | −5.017 | *** |
| Corporate governance → EPS | −1.035 | 0.429 | −2.413 | 0.016 |

Note: *** in row "*p* value" means "*p* value < 0.001".

The specific interpretation of the model is indicated below.

### 4.3.1. Interpretation of Coefficients between Observed Variables and Latent Variables

This part mainly explores whether various specific indicators of latent variables could fully represent them. From the path coefficient table, firstly, for external pressure, "proportion of state-owned shares" (1.99), "J-F coefficient" (1) and "loans" (2.13) passed the significance test at 0.01 level. This showed that, among sources of external pressure, "loans" contributed the most to external pressure, followed by "proportion of state-owned shares ", which measured the pressure of government supervision, and then the "J-F coefficient". Such results supported Hypothesis 1a, 1b and 1c, which meant that government supervision, public opinion represented by social media and loans could promote disclosure of corporations' environmental accounting information. Secondly, in regard to corporate performance, in addition to the "EPS" (1), among the rest of the observative variables only "Total asset turnover" (TAT) (0.59) approached the 0.1 level of significance test. However, as is mentioned later, "corporate performance" showed a positive correlation with both "disclosure level score" and "disclosure quality score" in 0.01 significance level. The reasons were that there are many indices in the financial evaluation system to measure all kinds of ability in a corporation, and one single index is usually unable to represent the whole performance of corporations. Only joint analysis with other indicators can comprehensively provide valuable evaluation results. For example, in 2001, China Securities Journal researched a performance evaluation model with Company Research Center, Tsinghua University. As many as 15 financial indicators were selected for the model, and they mainly derived from three aspects: profitability, solvency and growth. Thirdly, in regard to corporate governance, "Ownership concentration" (1) and "management shareholding ratio" (−0.49) passed the significance test at 0.01 level, indicating that, among the four indicators, "ownership concentration" could best represent the situation of corporate governance,

followed by "management shareholding ratio". The other observed variables did not pass any level of the significance test.

### 4.3.2. Interpretation of Results between Latent Variables and Explained Variables

As seen from Table 8, path coefficients between latent variable "external pressure" and observed variable "environmental accounting disclosure quality score" (hereinafter referred to as "quality score") and "environmental accounting disclosure level score" (hereinafter referred to as "level score") were 1.02 and 1.4, and the significance *p* values were 0.009 and 0.002, which meant they passed the significance test at 0.01 level. This meant that increase of external pressure of 1 unit would lead to the equidirectional change of disclosure quality score and disclosure level score of 1.02 and 1.4 units, respectively. Such results proved Hypothesis 1: External pressure would significantly positively affect the environmental accounting information disclosure of listed companies.

The path coefficients between "corporate performance" and "quality score" and "level score" were 2.87 and 3.82, while their significance coefficient *p* values were 0.001 and 0.005, passing the significance test at 0.01 level. This indicated that 1 unit change in corporate performance would lead to the equidirectional movement of disclosure quality score and disclosure level score of 2.87 and 3.82 units, respectively, which verified Hypothesis 2: Corporations with good performance will disclose a wider range of environmental accounting information with higher quality.

However, with regard to "corporate governance", the path significance coefficients of both "quality score" and "level score" did not pass the significance test at the level of 0.05. What is more, the path coefficient was also relatively small. Therefore, it was considered that "corporate governance" was not significantly associated with "quality score" and "level score". Differing from the above two results, this result meant that management governance would not influence corporations' environmental accounting information disclosure. Therefore, Hypothesis 3 "Corporate governance is positively correlated with corporate environmental accounting in-formation disclosure" was refuted.

### 4.3.3. Fit Degree Analysis and Hypothesis Testing

Model fitting analysis is one of the most important steps for AMOS, which is used to evaluate the degree of model matching with data. After inputting the data preprocessed by SPSS.26 into the above-mentioned model and performing the "calculate" operation, AMOS26.0 reported the fitting index table of the structural equation model, as shown in Tables 9 and 10 below.

**Table 9.** Summary of Various Fitting Indexes of Structural Equation Model.

| Model | NPAR | CMIN | DF | P | CMIN/DF | RMR |
|---|---|---|---|---|---|---|
| Default Model | 40 | 77.907 | 64 | 0.114 | 1.217 | 0.055 |
| | GFI | CFI | RMSEA | IFI | TLI | |
| | 0.949 | 0.969 | 0.033 | 0.971 | 0.956 | |

**Table 10.** Comparison of Fitting Degree Indexes of Structural Equation Models.

| Fitting Index | CMIN/DF | RMR | RMSEA | GFI | CFI | TLI | P |
|---|---|---|---|---|---|---|---|
| Adapt standard | (1, 3) | <0.08 | <0.05 | >0.9 | >0.9 | >0.9 | >0.05 |
| Model values | 1.217 | 0.055 | 0.033 | 0.949 | 0.969 | 0.956 | 0.114 |

CMIN represents the Chi-square value, while DF represents the degree of freedom of the model. The ratio CMIN/DF indicates whether the model is suitable for data. If the value is in the range of (1, 3), then it is considered to meet the standard. It can be seen that the value was 1.217, indicating that the model was suitable for analysis. RMR is the root mean square residual, which represents the square root of the sum of squares of residual after finding the difference between the actual matrix and the model matrix. The smaller

the value, the more suitable it is. In the chart, RMR = 0.055 < 0.08, which was in line with the standard. RMSEA is the approximate root mean square error, which represents the square root of the sum of squares of asymptotic residuals. Similar to RMR, its value obeys "the smaller the better" rule. As is shown in the chart, RMSEA = 0.033 < 0.05, which was in line with the standard. CFI, GFI and TLI are comparative fitting index, fitting degree index and non-standard fitting index, respectively, which were between 0 and 1. In the range of (0, 1), differing from the above-mentioned indices, the larger the value is, the better the fitting degree is. The CFI value of the model in this paper was 0.966 > 0.9, GFI value was 0.948 > 0.9, and TLI value was 0.952 > 0.9, all of which met the standard.

To sum up, the data generally conformed to the fitting indices of the model, and the fitting degree was fine.

### 4.4. Collinearity Diagnostics

To examine the possible collinearity problem among the above-mentioned variables, this paper used the variance inflation coefficient (VIF) test, which is one of the most commonly used mathematical methods, to verify the problem. The variance inflation coefficient is the ratio between the variance of the estimator of the regression coefficient and the variance when the independent variables are not linearly correlated. The specific calculation Formula (2):

$$\text{VIF} = \frac{1}{1 - R_i^2} \tag{2}$$

In the above formula, $R_i$ is the negative correlation coefficient between i variable and others variables in the regression analysis. The closer VIF is to 1, the less collinearity exists between variables. Tolerance is the inverse of VIF. We used SPSS. 26.0 to carry out the VIF test and the results are shown in Table 11.

As is shown in Table 11, the VIF value of none of the variables surpassed 3, which meant collinearity in the model was not significant.

**Table 11.** VIF/Tolerance Statistics of Observation Variable.

| Category | Observation Variable | VIF | Tolerance |
|---|---|---|---|
| External pressure | Loans | 1.384 | 0.723 |
| | J-F coefficient | 1.140 | 0.877 |
| | Proportion of state-owned equity (SOE) | 1.592 | 0.628 |
| | Whether audited by "Big Four" (Big Four) | 1.076 | 0.929 |
| Corporate performance | Earnings per share (EPS) | 1.160 | 0.862 |
| | Net profit growth rate (NPR) | 1.032 | 0.969 |
| | Total asset turnover (TAT) | 1.093 | 0.915 |
| | Debt financial leverage (DFL) | 1.075 | 0.930 |
| Corporate governance | Ownership concentration (Herf5) | 2.034 | 0.492 |
| | Management shareholdings (MH) | 1.296 | 0.771 |
| | Ratio of independent directors (RID) | 1.104 | 0.905 |
| | Degree of equity balance (ERR) | 1.557 | 0.642 |

### 4.5. Robustness Test

In the field of environmental accounting, most previous empirical research used regression analysis to explore factors affecting the environmental accounting information disclosure of listed companies or heavily polluting corporations. In this paper, as is illustrated above, the structural equation model was used, for the first time, to explore the relationship between the three latent variables of "external pressure", "corporate performance" and "corporate governance" and the two dimensions (i.e., quality and level of environmental accounting information disclosed by corporations). To test and verify the reliability of the conclusion, we adopted the method of regression analysis in a robustness test. At the same time, we constructed a new dependent variable "disclosure overall score" (hereinafter referred as "DOS") = "disclosure level the score" + "disclosure

quality score" to represent corporations' environmental accounting information disclosure situation in another way. After standardization, the regression analysis was conducted with 12 observation variables, respectively. While making regression between "DOS" and one of the observation variables in a latent variable, others were then set as control variables (for example, "loan", "proportion of state-owned shares" and "whether the four major audit" were set as control variables in the regression while analyzing "DOS" and "J-F coefficient"). The regression results are summarized in Table 12.

**Table 12.** Regression Coefficients of Robustness Test.

| Category | Observation Variable | Normalization Coefficient β | t-Statistic | Significance Level |
|---|---|---|---|---|
| External pressure | Borrowing costs (Loans) | 0.084 | 1.115 | 0.266 |
| | J-F coefficient | 0.144 * | 2.075 | 0.039 * |
| | Proportion of state-owned equity (SOE) | 0.239 *** | 3.244 | 0.001 |
| | Whether audited by "Big Four" (Big Four) | 0.098 | 1.451 | 0.148 |
| Corporate performance | Earnings per share (EPS) | 0.179 ** | 2.545 | 0.012 |
| | Net profit growth rate (NPR) | 0.097 | 1.413 | 0.159 |
| | Total asset turnover (TAT) | 0.182 *** | 2.628 | 0.009 |
| | Debt financial leverage (DFL) | 0.002 | 0.030 | 0.976 |
| Corporate governance | Ownership concentration (Herf5) | 0.213 *** | 2.671 | 0.008 |
| | Management shareholdings (MH) | −0.244 *** | −3.446 | 0.001 |
| | Ratio of independent directors (RID) | 0.053 | 0.772 | 0.441 |
| | Degree of equity balance (ERR) | 0.029 | 0.364 | 0.716 |

Note: *** represents significant at 0.01 level; ** represents significant at 0.05 level; * represents significant at the level of 0.1.

As is seen from the above table, the results of the robustness test generally supported the hypothesis verified by the structural equation model mentioned earlier.

### 4.6. Additional Finding

In the research process, we drew a scatter plot of the quality scores and level scores in the sample by using SPSS.26 software and found that there seemed to be certain linear relationships between the two parts (as shown in Figure 5).

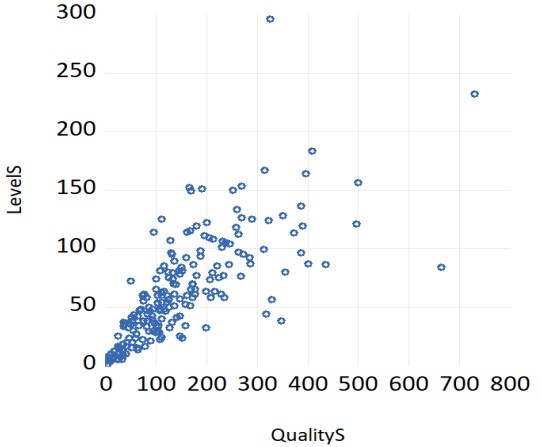

**Figure 5.** Scatter Diagram of Environmental Disclosure Quality Score and Disclosure Level. Note: the blue circle means the sample who owns corresponding level score and quality score.

Out of curiosity, we performed ordinary least squares regression (OLS) on Eviews.12 with Level Score ("LevelS" for short) as the explanatory variable and Quality Score ("QualityS" for short) as the explanatory variable (the regression results are shown in the Table 13).

**Table 13.** Ordinary Least Square Regression Coefficient.

| Variable | Coefficient | Std. Error | t-Statistic | Prob. |
|---|---|---|---|---|
| QualityS | 0.283428 | 0.018084 | 15.67305 | 0.000 |
| C | 20.34554 | 3.428498 | 5.934243 | 0.000 |
| R-squared | 0.553697 | Mean dependent var | | 62.300 |
| Adjusted R-squared | 0.551443 | S.D. dependent var | | 45.23429 |
| S.E. of regression | 30.29540 | Akaike info criterion | | 9.669818 |
| Sum squared resid | 181726.6 | Schwarz criterion | | 9.702802 |
| Log likelihood | −964.9818 | Hannan-Quinn criter. | | 9.683166 |
| F-statistic | 245.6446 | Durbin-Watson stat | | 1.930242 |
| Prob (F-statistic) | 0.0000 | | | |

Interpretation of results: The LevelS, QualityS, and the whole model passed the significance test at 0.01 level. Based on knowledge of econometrics, Durbin–Watson statistics (hereinafter referred to as D-W statistics) are the statistics that test whether there is autocorrelation between explanatory variables and explained variables, and has two auxiliary indices: DU and DL. If the D-W statistic falls within the (DU, 4-DU) interval, it is proved that there is no autocorrelation between them. According to D-W statistics critical value table, when sample size $n$ = 200 and explanatory variable K = 1, the corresponding DU value = 1.779. That is to say, no autocorrelation interval is (1.779, 2.221). From the above table, we obtained a D-W statistic = 1.930242, which was in the range. Therefore, there was no autocorrelation between quality score and level score.

As can be seen from the results table of the Eviews.12 ordinary least squares regression (OLS), if the quality score of environmental accounting disclosure is taken as the explanatory variable and the level score as the explained variable, the following regression model can be constructed to explain the relationship between the two:

$$\text{LevelS} = 20.34554 + 0.283428 \times \text{QualityS}.$$

According to the model, if the quality score increased 1QualityS, ceteris paribus, the level score would increase by about 0.283. In addition, the adjusted $R^2$ = 0.553 proved that the explanatory power of the model to the whole was 55.3%, which is relatively good in empirical studies.

Then we drew an interesting conclusion: there was a significant positive correlation between quality score of environmental disclosure and level score. This possibly showed that the disclosure of environmental accounting information by listed companies is a comprehensive consideration. If a listed company discloses environmental accounting information in a wide range, the quality of disclosure is highly likely to include accurate and quantitative information.

## 5. Conclusions

This article selected 200 listed companies from the 2015–2019 panel data to explore external pressure, corporate performance, corporate governance (latent variables) and the sample companies' environmental accounting information disclosure by using a structural equation model. Then we used regression analysis to examine 12 observed variables (which belonged to the aforementioned three latent variables) in the robustness test. Overall, we drew the following conclusions:

Firstly, consistent with the reputation theory model, external pressure could significantly positively affect the quality and level of environmental accounting information disclosure of listed companies. Among many sources of external pressure, whether audited by the "Big Four" (Big Four) did not show any effect. What really prompted corporations to improve their environmental accounting information were government supervision (SOE), media pressure (J-F coefficient) and bank pressure (loans).

Secondly, corporate performance, as a whole, could significantly positively affect the quality and level of disclosure, which verified the signal transmission theory. Earnings per share (EPS) and total assets turnover (TAT), which reflect the profitability and sales ability of corporations, are crucial factors. The growth rate of net profit (NPR) and debt financial leverage (DFL) has no obvious effect on it.

Finally, corporate governance had no significant impact on EAID. It was found that management shareholding (MH) was negatively correlated with disclosure, while ownership concentration (Herf5) was positively correlated with disclosure. We found that highly "centralized" companies usually paid more attention to disclosure of information concerning their social responsibilities, which differed from some previous studies. At the same time, the study did not find any impact from the proportion of independent directors (RID) and the degree of equity balance (ERR) on both quality and level of environmental accounting information disclosure.

Based on the above research, we found that external pressure, especially government supervision and media pressure, had a particularly significant impact on corporate environmental accounting information disclosure. Therefore, relevant environmental protection departments of the government could link their regulatory deterrence with media opinion to "reward the good and punish the bad", such as establishing some corporations that do well in environmental accounting information disclosure as "model corporations", give them substantial rewards like tax relief, etc. [50]. and publicize them positively through the public media. At the same time, the government should also increase punishment intensity for companies who perform poorly in this field, especially those who are "able but not willing". The dual pressure of "government supervision + media opinion" combine external pressure efficiently, thus helping corporations standardize and improve their disclosure of environmental accounting information.

It is worth noting that while exploring corporate governance, we found that, although it has little impact on EAID, the proportion of management shareholding was significantly negatively correlated with it. This might be due to the fact that China's environmental accounting did not establish a certain system for exclusive laws and regulations of environmental accounting, which led to the disclosure of environmental accounting information becoming a kind of voluntary disclosure. Some corporations seized this characteristic and nurtured the idea that "it is better to be silent than to speak more but lose more". Less disclosure of this information could not only avoid "getting into trouble", but also reduce the costs of disclosure. When executive compensation is closely related to target profit, management's manipulation of calculated profit would be more significant [51]. Therefore, reducing a kind of "voluntary disclosure cost" is likely to become a kind of "profit manipulation" by management. To some extent, this suggests that government department laws and regulations related to environmental accounting are urgently needed to avoid more corporations saving this "voluntary disclose cost" as much as they can.

It is gratifying that China's Environmental Protection Law was revised and officially implemented on 1 January 2015 (this was also the reason for the sample selection time period in the paper). We believe that more laws and regulations will be issued in the future to regulate China's information disclosure about the performance of social responsibility of listed companies.

## 6. Limitations and Future Research Directions

Finally, we think there is room for further improvement in corporate performance. Recent studies found that there are many potential and interesting factors that may influence corporate governance indirectly, such as internal control, early life experiences of the board and reporting location of some financial indicators [52–54]. As for auditing, on the condition key audit matters (KAM) reduce information acquisition cost [16,55], the question is: Does this mean that EAID may make it easier to gain some less accessible information if environmental accounting information becomes one of the KAMs? The additional conclusion that "environmental accounting disclosure quality score and disclosure

level score were significantly positively related to relationship" raises the possibility that, in order to save unnecessary auditing costs, when auditing corporate social responsibility (CSR) information examining one dimension of corporate environmental accounting information (such as disclosure quality), might allow accounting firms to acquire basic acknowledgement of another dimension (such as disclosure level). Whether or not this is the case may be a possible direction for further research.

**Author Contributions:** Conceptualization, M.J. and Y.J.; methodology, Y.J.; software, Y.J.; validation, Y.J.; formal analysis, Y.J.; investigation, Y.J.; resources, S.D. and M.J.; data curation, Y.J.; writing—original draft preparation, Y.J. and M.J.; writing—review and editing, S.D.; visualization, Y.J.; supervision, Y.J.; funding acquisition, S.D. All authors have read and agreed to the published version of the manuscript.

**Funding:** This research was funded by the "Annual project of Philosophy and Social Science Research Planning of Zhejiang Province", funding number "20NDJC154YB" (funder: Zhejiang Philosophy and Social Science Development Planning Office) and the "Graduate Innovation Fund of Jilin University", funding number "2022143" (funder: Jilin University).

**Institutional Review Board Statement:** Not applicable.

**Informed Consent Statement:** Not applicable.

**Data Availability Statement:** Primary data for this article is available in the website: https://figshare.com/articles/dataset/Primary_data_for_Who_makes_corporations_tell_more_about_CSR_-Research_on_the_influencing_factors_of_corporate_environmental_accounting_information_disclosure_sav/21058450 (accessed on 24 November 2022).

**Acknowledgments:** Our team sincerely appreciate the help provided the fund, "Annual Planning Project of Philosophy and Social Science Research of Zhejiang Province". Without their subsidization, it would have been more difficult to get primary data from the CSMAR database.

**Conflicts of Interest:** The authors declare no conflict of interest.

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
