# Peer review of "Environmental Accounting Information Disclosure Driving Factors: The Case of Listed Firms in China"

_sustainability, doi:10.3390/su142315797_

Round 1

Reviewer 1 Report (Previous Reviewer 1)

The author has improved the study and now it looks fine. However, see H1 & H1b, it is better to make hypothesis in present tense instead of future tense. figure 6 is just copied from E-View which looks bad, try to use academic formitting in word. moreover, references also need attention. 

Author Response

Thanks for your suggestions. Here are my response to your suggestions point by point. Revision paper is attached.

Point 1: It is better to make hypothesis in present tense instead of future tense.

Response 1: Done. As you suggested, all of our hypothesis in this paper have been changed to present tense. Thanks for your suggestion and I will keep it in mind in my future researches.

Point 2: Figure 6 is just copied from E-View which looks bad, try to use academic formitting in word.

Response 2: Done. As you can see in the revision, Figure 6 has been changed to Table 13 in an academic format of Word. In my future research, I will pay more attention to the format of my paper.

Point 3: References also need attention.

Response 3: Done. In the revision, we have unified the format of references according to the format requirement of Sustainability.

Epilogue: Sincerely thanks for your valuable suggestions to our paper. These suggestions have improved our research greatly. I will remember these suggestions in my future researches!

Reviewer 2 Report (Previous Reviewer 2)

The authors have adequately addressed all the amendments required. Thus, the paper can be published in present form.

Author Response

Response:

Some formats and reference style have also been improved in the revision. Sincerely thanks for your valuable suggestions to our paper. These suggestions have improved our research greatly. I will remember these suggestions in my future researches!

Reviewer 3 Report (Previous Reviewer 3)

Thank you for editing the article according to my comments. I think the article is now ok and can be published.

Author Response

Response:

Some formats and reference style have also been improved in the revision. Sincerely thanks for your valuable suggestions to our paper. These suggestions have improved our research greatly. I will remember these suggestions in my future researches!

This manuscript is a resubmission of an earlier submission. The following is a list of the peer review reports and author responses from that submission.

Round 1

Reviewer 1 Report

The paper has many good points but needs improvement. 

1) The introduction is little bit lengthy and needs improvement with current relevant references. There is too much quotation. Introduction failed to establish the need of the study.

2)  Literature review is too lengthy and not created the interest of reader. 

3) method is good and detailed

4) Research implications and future recommendations are not presented satisfactorily. I suggest the authors to read some scientific papers published in journals and improve/revise their study usefulness for industry and literature.

5) Language use and communication are not acceptable, copy editing/ proofreading by a native English proof-reader will enhance the communication quality of the manuscript.

2

Reviewer 2 Report

The paper entitled “Whio makes corporates tell more about CSR? - Research on the influencing factors of environmental information disclosure” aims at finding and analyzing the factors connected with environmental information disclosure by firms. This is a contemporary and interesting subject to analyze and closely connected with the Journal’s aims and scope.

There are several amendments to be addressed before the paper can be published.

1. The authors must carefully review the title and try to point out both the paper’s aim and area of analysis. Thus a title like the following one is proposed: “Environmental Information Disclosure Influencing Factors: The Case of Listed Firms in China”

2. The paper’s abstract should be radically re-written. The authors must keep in mind that a paper’s abstract must include the following parts: purpose, design/methodology/approach, findings, originality/value.

3. Based on the paper’s introduction it can be derived that the paper’s aim is to explore the factors affecting enterprise environmental accounting information disclosure. However, the introduction does not support at all this aim. It is necessary to add more information why environmental accounting information disclosure is important and why the case of China is a relevant and important case to analyze. The rationale (both theoretical and organizational) of the research should be pointed out.

4. The paper’s literature review lacks of information about how the environmental protection can be connected with a firm’s performance. This is an important subject to be analyzed because that is the base of how to address the environmental accounting information disclosure. To do so, the following papers can be used:

- Skordoulis, M., Kyriakopoulos, G., Ntanos, S., Galatsidas, S., Arabatzis, G., Chalikias, M., & Kalantonis, P. (2022). The Mediating Role of Firm Strategy in the Relationship between Green Entrepreneurship, Green Innovation, and Competitive Advantage: The Case of Medium and Large-Sized Firms in Greece. Sustainability, 14(6), 3286.

- Rehman, S. U., Kraus, S., Shah, S. A., Khanin, D., & Mahto, R. V. (2021). Analyzing the relationship between green innovation and environmental performance in large manufacturing firms. Technological Forecasting and Social Change, 163, 120481.

- Duanmu, J. L., Bu, M., & Pittman, R. (2018). Does market competition dampen environmental performance? Evidence from China. Strategic Management Journal, 39(11), 3006-3030.

- Skordoulis, M., Ntanos, S., Kyriakopoulos, G. L., Arabatzis, G., Galatsidas, S., & Chalikias, M. (2020). Environmental innovation, open innovation dynamics and competitive advantage of medium and large-sized firms. Journal of Open Innovation: Technology, Market, and Complexity, 6(4), 195.

5. Moreover, as far as the literature review is concerned, it is strongly proposed that the sections 2 and 3 are merged into one section. At the same time the sub-section 2.4 about literature summary must be deleted and its paragraphs must be integrated in the paper’s introduction and conclusion sections.

6. In the paper’s methodology section the authors must explain the remedies they adopted to address the possibility of a double causal relationship which is considered high based on the research variables’ description.

7. Descriptive statistics must be moved from paper’s methodology section to paper’s results section.

8. Section 6 must be renamed to “limitations and future research directions”. It is very important to some paragraphs about the paper’s possible limitations and caveats.

9. Last, the paper has several grammar flaws. It is strongly recommended to be proofread by a native speaker of English language or a language agency.  

Reviewer 3 Report

I appreciate the topicality and necessity of the topic. The article is consistent, the individual parts follow each other perfectly, the individual parts are described in detail. I also appreciate the development of hypotheses and used methods based on literature analysis. It's also good that you described your sample selection, why you didn't include more recent data, and why you included these particular companies. Furthermore, the description of the editing and selection of data is also very good, but the results and conclusions achieved are also beneficial. I only have small notes for the authors, otherwise I recommend publishing the article:

·         Typo already in the title of the article.

·         Some of the literature references used are quite outdated, although there should be more recent studies that address the topic. In some cases this can be understood, these are innovative, original ideas and models, somewhere it seems to me that more recent sources could be cited. Please try to double check.

·         I don't really understand the note under figure 1 or figure 2 "Already got the citing right".

·         The confirmation or refutation of the hypotheses that were established should be better described in the results section.

·         Description in section “5. Conclusion' should be in the past tense, it is better to use the past tense when summarizing the results you have done than the present tense.